# Combining Radiotherapy and Immunotherapy in Head and Neck Cancer

**DOI:** 10.3390/biomedicines11082097

**Published:** 2023-07-25

**Authors:** Juliana Runnels, Julie R. Bloom, Kristin Hsieh, Daniel R. Dickstein, Yuhao Shi, Brianna M. Jones, Eric J. Lehrer, Richard L. Bakst

**Affiliations:** 1Department of Radiation Oncology, Icahn School of Medicine at Mount Sinai, New York, NY 10029, USAkristin.hsieh@mountsinai.org (K.H.); daniel.dickstein@mountsinai.org (D.R.D.); eric.lehrer@mountsinai.org (E.J.L.); richard.bakst@mountsinai.org (R.L.B.); 2Roswell Park Comprehensive Cancer Center, Buffalo, NY 14263, USA; yshi24@buffalo.edu

**Keywords:** immunotherapy, radiation therapy, head and neck cancer squamous cell cancer

## Abstract

Head and neck squamous cell carcinoma (HNSCC) is a leading cause of morbidity and mortality globally. Despite significant advances in well-established treatment techniques, prognosis for advanced-stage HNSCC remains poor. Recent, accumulating evidence supports a role for immunotherapy in HNSCC treatment. Radiation therapy (RT), a standard treatment option for HNSCC, has immunomodulatory and immunostimulatory effects that may enhance the efficacy of immunotherapy. In several cancer types, combining RT and immunotherapy has been shown to improve tumor response rates, increase survival, and reduce toxicity compared to traditional chemotherapy and radiation therapy. This review provides a timely overview of the current knowledge on the use of RT and immunotherapy for treating HNSCC. It highlights the potential advantages of combining these therapies, such as improved tumor response rates, increased survival, and reduced toxicity. The review also discusses the challenges that need to be addressed when redefining the standard of care in HNSCC, and proposes further research to optimize treatment combinations, minimize radiation-induced toxicity, and identify suitable patient populations for treatment.

## 1. Introduction

Head and neck squamous cell carcinomas (HNSCC) continue to be a leading cause of morbidity and mortality worldwide, with over 700,000 new cases annually and 350,000 deaths globally [1]. Roughly 60% of cases at time of diagnosis are classified as locally advanced. Despite evolution of oncologic care, including more precise surgical techniques, radiation therapy and advancement in systemic therapy, the 5-year overall survival rate of patients with HNSCC has not changed significantly. Locally advanced (LA) HNSCC patients have relapse rates of about 50% within the first two years [2,3,4]. While some subsets of disease, such as human papilloma virus (HPV)-mediated oropharyngeal cancer, have high 3-year survival rates of over 90% [5], other sites such as LA hypopharyngeal cancer have a 5-year disease control rate of less than 30% [6].

The subsite of disease, stage, histology, and patient comorbidities dictate appropriate treatment recommendations. LA disease is often treated with multimodality therapy. For patients with locoregionally advanced disease, definitive concurrent chemoradiation therapy versus surgical resection with adjuvant radiation therapy with or without chemotherapy is the standard of care. Radiation therapy is commonly delivered via intensity-modulated radiation therapy (IMRT) using contemporary computer-based planning and radiation delivery with or without simultaneous integrated boost (SIB) or dose painting, assigning different dose levels to different anatomic areas. If a patient undergoes surgical resection, adjuvant radiation therapy with or without concurrent cisplatin chemotherapy is recommended if adverse pathologic features are noted [7,8,9,10]. Relapsed/refractory disease is often treated with an attempt to cure with surgery or radiation, followed by systemic therapy, when feasible. A second course of radiation therapy may be considered, depending on the time interval from the first course of radiation therapy, with use of intensity-modulated particle therapy, if available. In the setting of oligometastatic disease, curative therapy with radiation therapy or surgery may be appropriate, depending on the patient’s performance status, prior therapy, interval since prior therapy, and desire for functional preservation [11,12,13].

Over the last two decades, there has been a dramatic increase in the interest in and development of immunotherapy as an oncologic weapon. Some types of cancer cells are immunogenic, allowing them to constantly mutate with select clones, developing the ability to evade immune-mediated recognition and destruction [14,15]. Several targets including immune checkpoint inhibitors (ICI) are of significance for their ability to block checkpoint proteins from binding to their partner proteins. Immune checkpoints keep immunologic cells such as T-cells in a dysfunctional state; ICIs prevent the propagation of the “off” signal, allowing space for T-cells to kill the cancer cells.

Two examples of these targets are the programmed cell death protein 1 (PD-1) and cytotoxic T-lymphocyte-associated protein 4 (CTLA-4). When PD-1 on T-cells binds to programmed death-ligand 1 (PD-L1) on cancer cells, T-cells are unable to kill other cells, including cancer cells. Anti-PD1 and anti-PD-L1 therapy interrupt PD-1 engagement via its ligand, allowing for the continued activation of T-cells and immune-mediated anti-cancer response [16]. Similarly, CTLA-4 on T-cells can bind to a protein on antigen presenting cells, B7. Introducing an anti-CLLA-4 antibody to the system blocks the binding of CTLA-4 and B7, resulting in the increased ability of T-cells to kill cancer cells.

Immune checkpoint inhibitors targeting PD-1/PD-L1 complex have shown promising results in several cancer types including melanoma, renal cell carcinoma, non-small cell lung cancer (NSCLC), and head and neck cancer [17]. Two trials, KEYNOTE-040 and CheckMate 141, looked at the immunotherapies pembrolizumab and nivolumab, respectively, for treatment of recurrent or metastatic HNSCC after disease progression on or after platinum-based therapy. Both trials found an improvement in overall survival (OS) and led to the approval of pembrolizumab and nivolumab for the treatment of recurrent or metastatic HNSCC after disease progression on/after platinum therapy. In KEYNOTE-040, the median OS was 8.4 months with pembrolizumab and 6.9 months with the standard of care comprising methotrexate, docetaxel, and cetuximab [18]. In CheckMate 141, the median OS was 9.5 months with nivolumab and 6.2 months with the investigator’s choice of therapy [19]. The KEYNOTE-048 trial evaluated the use of ICI pembrolizumab, an anti-PD1 antibody, as a first-line treatment for recurrent or metastatic HNSCC considered incurable with local therapies [20]. PD-L1 expression was assessed in all subjects and characterized as a PD-L1 combined positive score (CPS). Subjects were randomized to one of three arms—pembrolizumab alone, pembrolizumab with chemotherapy, or cetuximab with chemotherapy—irrespective of PD-L1 status. Patients who received pembrolizumab with a platinum chemotherapy and 5-FU vs. cetuximab with chemotherapy had an improved median overall survival (OS) of 13.0 vs. 10.7 months. For pembrolizumab monotherapy, greater PD-L1 expression was associated with greater response. Pembrolizumab monotherapy was associated with a significant overall survival benefit in patients with a PD-L1 CPS of 20 or more or 1 or more, and had non-inferior OS in the overall cohort compared to cetuximab with chemotherapy. Pembrolizumab given with chemotherapy also significantly improved OS in the populations with PD-L1 CPS greater than 20, CPS greater than 1, and in the total population, compared with cetuximab and chemotherapy. Based on this trial, pembrolizumab was approved for use as a first-line treatment for HNSCC patients with recurrent, unresectable, or metastatic disease.

Radiation therapy may enhance the efficacy of immunotherapy through its immunomodulatory and immunostimulatory effects. The abscopal effect refers to a phenomenon observed in radiation therapy wherein localized radiation treatment not only affects the targeted tumor, but also triggers a systemic immune response, leading to the regression or control of distant, untreated tumors. It occurs when radiation-induced cell death releases tumor-specific antigens and danger signals, which activate immune cells, such as T cells and natural killer cells, to recognize and attack cancer cells throughout the body. Radiation can also induce in situ vaccination by triggering the killing of tumor cells and generating a systemic immune response [21]. Ionizing radiation induces chemokines and stimulates the recruitment of effector T-cells, increasing effector cell response [22]. Some experiments suggest that the use of hypofractionation (10–24 Gy in a single fraction), may lead to a larger release of immunogenic tumor antigens, suggesting the immunomodulatory effect may be more notable in this treatment setting [23]. The combination of immunotherapy via CTLA-4 blockage with radiation therapy has been found to be synergistic and therapeutically effective in animal models in comparison to either modality alone [24,25,26].

Given the discrete molecular landscapes between HPV-positive and HPV-negative HNSCC, several studies have looked at the correlation between HPV status and the effect of immunotherapy [27]. Expression of the HPV-associated E6 and E7 oncogenes inactivates the tumor suppressor proteins p53 and pRb, respectively, which are frequently mutated in HPV-related mucosal squamous cell carcinomas. Inactivation of p53 and pRb promotes genomic instability via the EGFR and PI3k pathways and the development of cancer, and is responsible for the upregulation of p16 protein expression [28,29]. There are conflicting results for incorporating ICIs in HNSCC clinical trials. One systematic review and meta-analysis concluded that HPV-positive HNSCC patients who received PD-1 or PD-L1 therapy demonstrated improved OS compared to HPV-negative patients [30]. This is in accordance with survival trends of these two populations after chemoradiation therapy [31,32]. On the contrary, another systematic review and meta-analysis demonstrated that PD-L1 expression enhanced the response to immunotherapy, and HPV status did not affect tumor response or overall survival when patients received anti-PD-L1 therapy [33].

The aim of this narrative review is to synthesize the current literature to date regarding the use of immunotherapy with radiation therapy in the treatment of HNSCC. This manuscript highlights current ongoing trials for LA HNSCC and recurrent or metastatic HNSCC, and postulates future directions within this field.

## 2. Radioimmunotherapy for LA HNSCC

The potential role of radioimmunotherapy in the locally advanced (LA) setting can be generally classified into (i) the addition of IO to definitive chemo-RT/RT, or (ii) IO + RT in perioperative (neoadjuvant and adjuvant) settings. The IO agents under investigation as monotherapy or in combination with immunotherapy regiments include drugs targeting PD-1, PD-L1, and CTLA-4. Disappointingly, initial results from phase II/III studies assessing definitive IO + RT have been largely negative. For example, NRG-HN004 evaluated the efficacy of concurrent and adjuvant durvalumab compared to standard RT with cetuximab. RT with durvalumab led to significantly worse locoregional failure (LRF) and did not signal improved progression-free survival (PFS) in patients with a contraindication to cisplatin [34].

However, ongoing trials in the neoadjuvant setting supported by pre-clinical evidence may help define a new role for IO agents in LA HNSCC. Below, we highlight initial results from major phase III studies and ongoing trials that may identify new approaches to incorporate radioimmunotherapy in this setting (Table 1).

### 2.1. Radiation and Immunotherapy in the Definitive Setting

Phase III trials testing addition of IO to definitive chemo-RT/RT regimens in LA HNSCC include JAVELIN Head and Neck 100, GORTEC-REACH, NRG-HN005, KEYNOTE-412, and EA3161. Trials are summarized in Table 1.

JAVELIN Head and Neck 100 is a trial testing addition of 12-month avelumab versus placebo in 697 patients with LA HNSCC that received cisplatin-based chemo-RT. Initial results showed an HR of 1.21 (95% CI, 0.93–1.57) and 1.31 (95% CI, 0.93–1.85) for progression-free survival (PFS) and OS, respectively. The trial was prematurely discontinued because the boundary for futility had been crossed. On subgroup analysis, patients with tumors that had PD-L1 high expression (CPS ≤ 25%) showed a trend of potential benefit, with an HR of 0.59 (95% CI, 0.28–1.22) [35].

NRG-HN005 aims to evaluate the effectiveness and safety of de-intensified radiation therapy in combination with either chemotherapy using cisplatin or immunotherapy using nivolumab in treating patients with early-stage, HPV-positive, non-smoking associated oropharyngeal cancer. The study will compare the two treatments’ efficacies in terms of PFS and OS, as well as their safety profile. The trial will provide valuable insights into the potential of de-intensified radiation therapy combined with chemotherapy or immunotherapy as a treatment option for patients with oropharyngeal cancer [37].

GORTEC-REACH is a trial involving two patient cohorts: cisplatin-fit and cisplatin-unfit. The trial assessed weekly cetuximab and avelumab concurrent with radiation followed by avelumab for 12 months in both cohorts versus the standard of care. The standard of care for the cisplatin-fit cohort was cisplatin-based chemoradiation, and for the cisplatin-unfit cohort was cetuximab and radiation. Radiation therapy was delivered at a dose of 70 Gy over 6.5 weeks. The primary endpoint of PFS was not met for either cohort. For the cisplatin-fit cohort, PFS HR was 1.27 (95% CI, 0.83–1.93), with a futility boundary for efficacy crossed favoring the standard of care, cisplatin. For the cisplatin-unfit cohort, PFS HR at 2 years was 0.85 (*p* = 0.15). [36]

KEYNOTE 412 is a trial examining the efficacy of adding concurrent and adjuvant pembrolizumab to cisplatin-based chemoradiation versus placebo. The trial enrolled 804 patients, and on final analysis for primary end point (efficacy boundary survival), the results showed a favorable trend, but no statistically significant benefit for pembrolizumab arm HR 0.83 (05% CI, 0.68–1.03). A subset analysis of PD-L1 positive (CPS ≥ 1) patients at 36 months showed an EFS of 58% in the pembrolizumab arm vs. 51.8% in the control arm; overall survival was 71.4% vs. 70.2%, respectively. In the PD-L1 high-expression (CPS ≥ 20) subgroup, neither median EFS or OS were reached in either arm [38].

EA3161 is a trial testing whether maintenance immunotherapy (nivolumab) following definitive treatment with radiation and chemotherapy (cisplatin) results in significant improvement in OS and PFS for patients with intermediate-risk HPV-positive oropharynx cancer that has spread to nearby tissue or lymph nodes [39].

Together, these trials show some potential benefits in specific patient subgroups including PD-L1 positive patients. Ongoing trials NRG-HN005 and EA3161 aim to evaluate outcomes associated with de-intensification of radiation therapy and addition of nivolumab for HPV-positive oropharynx cancer.

### 2.2. Radiation and Immunotherapy in the Adjuvant Setting

Several phase III trials are currently ongoing to test adjuvant radioimmunotherapy following definitive local/regional surgery. Trials are summarized in Table 2. Amongst these, IMvoke010 assesses adjuvant atezolizumab following definitive local therapy that can involve both surgical (induction chemotherapy/surgery, induction chemotherapy/surgery/RT) and non-surgical (induction chemo/RT or concurrent chemo/RT) approaches [40]. Other trials such as NIVOPOSTOP and RTOG1216 focus on testing the addition of IO (nivolumab, durvalumab, atezolizumab) to the standard-of-care adjuvant regimen of cisplatin/RT to post-surgical patients only [41].

### 2.3. Radiation and Immunotherapy in the Neoadjuvant Setting

The advantages of the use of IO in the neoadjuvant setting have been previously reviewed [43,44]; these include (i) the opportunity to “debulk” tumors prior to surgical resection, (ii) giving treatment when there are higher levels of tumor antigens with gross disease still intact, and (iii) the opportunity to assess pathological response to treatment in the resected tissue. Furthermore, a preclinical study using syngeneic murine oral cavity carcinoma models showed that neoadjuvant, not adjuvant, anti-PD-1 treatment is capable of inducing anti-tumor immunologic memory responses [45]. Trials are summarized in Table 3.

Clinically, several phase III studies have begun to assess IO for LA HNSCC in the neoadjuvant setting. KEYNOTE-689 (MK-3475-689) is a trial assessing neoadjuvant pembrolizumab prior to surgery with standard-of-care adjuvant treatment and continued pembrolizumab for LA HPV-negative HNSCC. Patients in this trial received two doses of neoadjuvant pembolizumab, which notably were able to induce higher levels of pathological response versus one dose in initial phase II studies [46]. The second phase III study is IMSTAR-HN, which examines neoadjuvant nivolumab followed by surgery, then adjuvant chemoradiation with (i) nivolumab or (ii) combination nivolumab and ipilimumab, versus standard-of-care surgery followed by adjuvant chemoradiation [47]. Lastly, CompARE aims to examine treatment escalation for intermediate/high risk oropharyngeal cancer. This will involve control arm 1: control chemoradiation (70 Gy in 35 Fractions + Cisplatin); arm 2: dose-escalated chemoradiation (64 Gy in 25 Fractions + Cisplatin); arm 3: Durvalumab + chemoradiation (70 Gy in 35 Fractions + cisplatin). Dose-escalated chemoradiation describes the delivery of a higher dose of RT per daily fraction. Across all treatment arms, patients will be assessed clinically and radiographically 3 months post-treatment, and may undergo neck dissection if indicated as per the current international gold standard. Initial results from these studies have yet to be reported [48].

IO for LA HNSCC clearly offers several advantages, including the administration of treatment during a phase of higher tumor antigen levels with intact gross disease and the ability to assess the pathological response to treatment in resected tissue. Clinical trials are ongoing to assess the benefits of neoadjuvant IO followed by surgery and adjuvant treatment, as well as treatment escalation in specific patient populations. Although initial results are pending, these studies hold promise in advancing the neoadjuvant treatment approach for LA HNSCC, and may provide valuable insights into optimizing patient outcomes.

## 3. Radioimmunotherapy for Recurrent and Metastatic HNSCC

For the treatment of recurrent and metastatic cancer, the combination of immunotherapy and radiotherapy is being actively investigated. Previous and ongoing studies are investigating the synergistic and therapeutic effects of this treatment combination. One preclinical study investigated the immune-mediated mechanism behind the abscopal effect, which refers to the reduction in tumor growth outside the area targeted by radiation. They found that enhancing the number of dendritic cells, a type of immune cell, using Flt3-Ligand (Flt3-L), in combination with radiation therapy, resulted in impaired growth of both the irradiated tumor and the non-irradiated tumor in mice. This effect was shown to be tumor-specific and dependent on the presence of T cells, suggesting the involvement of the immune system in mediating the abscopal effect induced by radiation [49]. Due to immune stimulation against tumor cells by directed radiation, immunotherapy will target both radiated and non-irradiated lesions systemically [50].

For metastatic disease, this treatment combination has been shown to be well tolerated. A phase I trial investigating SBRT and pembrolizumab to treat metastatic cancer illustrated that patients tolerate the combination with approximately 9.7% of patients experiencing grade 3 colitis, hepatitis, and pneumonitis [51]. Similarly, a phase II prospective trial investigating SBRT with and without durvalumab for the treatment of metastatic HNSCC illustrated that toxicities were similar between the two arms, with no significant difference, illustrating that the potential addition of immunotherapy to radiation is likely safe [52].

There is conflicting evidence as to the benefit of combining immunotherapy and radiation for the treatment of metastatic HNSCC. A phase II prospective clinical trial illustrated that SBRT combined with PD-1 receptor inhibitor pembrolizumab improved the overall response rate (ORR) in patients with NSCLC [53]. However, a phase II trial investigating the combination of another PD-1 receptor inhibitor, nivolumab, with or without SBRT, did not show any improvement in the ORR, OS, or PFS in the treatment of metastatic HNSCC [52]. The researchers concluded that there was no evidence to support the addition of nivolumab to SBRT for metastatic HNSCC from this study, and further research is warranted to understand the benefit of combination immunotherapy and radiotherapy in the treatment of metastatic HNSCC. More studies are needed to understand the combination of immunotherapy and radiation for the treatment of recurrent head and neck cancers. 

## 4. Discussion

The use of both radiotherapy and immunotherapy has shown great promise as a treatment strategy for patients with LA and recurrent/metastatic HNSCC. Several studies have explored the combined synergistic and therapeutic effects of these treatments, providing various mechanistic rationales for their effectiveness. Based on several phase II/III clinical trials, it appears that adding immunotherapy to radiation is a safe option. 

Overall, further research is needed to optimize the use of radioimmunotherapy for HNSCC, and to identify the patients who are most likely to benefit from this treatment. Ongoing trials are exploring various immunotherapeutic agents in order to determine the optimal timing, dose, sequence, and therapeutic agent for the combination of RT and immunotherapy in HNSCC treatment. Further research is needed to establish the optimal dose, fractionation, target volume, and field size for RT combined with immunotherapy.

Optimal immunotherapeutic agent selection and sequencing with RT are active areas of investigation. Results from several ongoing clinical trials could guide these decisions for the treatment of patients with HNSCC. Patients with HPV-LA HNSCC are receiving neoadjuvant durvalumab and concurrent SBRT prior to surgical resection in the phase I trial NCT03635164 [54]. Patients with HPV + LA HNSCC are receiving SBRT and durvalumab with or without tremelimumab prior to surgical resection in the phase I trial NCT03618134 [55]. In NCT03522584, patients sequentially receive durvalumab, hypofractionated radiation therapy, and tremelimumab [56]. ECOG EA3191 is a phase II trial studying the effect of pembrolizumab in combination with radiation therapy or pembrolizumab alone compared to the standard approach of chemoRT after surgery in patients with recurrent HNSCC [57]. Camrelizumab with SBRT and camrelizumab alone are being compared in a phase II trial of patients with R/M HNSCC [58].

For both metastatic and recurrent HNSCC, there are several ongoing clinical trials investigating radiotherapy techniques. Immunotherapy and brachytherapy are being studied for patients with recurrent HNSCC in NCT04340258. Perioperative pembrolizumab and Cesium-131 will be administered to patients prior to salvage surgery in this phase I/II trial. Cesium-131, a low-energy gamma isotope, will be placed at the time of surgery, and will deliver 60–70 Gy to the area of disease [59].

De-escalation has been a major theme in HNSCC RT. This shift aims to achieve effective cancer treatment while minimizing the potential toxicity associated with traditional high-dose radiation regimens. The radiobiological rationale for de-escalation stems from a better understanding of tumor biology and the dose–response relationship. Studies have shown that certain tumors may exhibit a phenomenon called the “dose–response plateau,” where further increasing the radiation dose beyond a certain threshold may not significantly improve local control. Additionally, normal tissues surrounding the tumor have a finite tolerance to radiation, beyond which the risk of toxicity increases. By de-escalating radiation therapy, it is possible to achieve a balance between effective tumor control and minimizing the risk of treatment-related side effects.

Several clinical trials have provided valuable insights into the efficacy and safety of de-escalation strategies. The RTOG 9003 trial, for instance, evaluated the impact of decreasing radiation dose in patients with human papillomavirus (HPV)-positive oropharyngeal squamous cell carcinoma. The trial demonstrated that reducing radiation dose while concurrently administering chemotherapy yielded similar outcomes compared to standard-dose radiation therapy, while reducing treatment-related toxicity. To achieve de-escalation, alternative fractionation and hypofractionation techniques have gained prominence. These approaches involve delivering higher radiation doses per fraction over a shorter treatment period, thereby reducing the overall treatment duration. QUADSHOT is a radiation therapy technique that allows for 14 Gy to be given in four fractions, given twice daily, at least 6 h apart, over two days [60]. QUADSHOT is being studied in combination with pembrolizumab in NCT04454489 [61]. HNSALV (NCT04754321) is a phase I trial examining possible side effects of pembrolizumab and radiation therapy given before and during surgery for patients with recurrent HNSCC. Patients receive pembrolizumab and QUADSHOT (EBRT delivered over two fractions in two days) prior to salvage surgery. Following surgery, pembrolizumab is continued, and they subsequently undergo intraoperative radiation therapy (IORT) [62]. Continued research and long-term follow-up are crucial to refine de-escalation protocols, identify optimal patient selection criteria, and determine whether modifications in dose fractionation modulate the effects of radioimmunotherapy treatment.

The toxicity from RT for head and neck cancer is dose-dependent. As such, target dose and volume reduction are important potential strategies to mitigate toxicity. The REWRITe trial (GORTEC 2018-02) is investigating treatment of LA HNSCC with durvalumab and RT without prophylactic neck irradiation. RT is given at a dose of 69.96 Gy in 33 fractions to the macroscopic tumor volume, and a second optional dose is given to the immediately adjacent nodal level, at a dose of 52.8 Gy, in 33 fractions. 

There is also a paucity of data to guide patient selection for radioimmunotherapy regimens. The location, stage, and extent of the tumor ought to play a crucial role in patient selection. Radioimmunotherapy may be considered for patients with LA or recurrent HNSCC who have not responded adequately to standard therapies or have limited treatment options. The patient’s overall health and functional status are important considerations. Biomarkers are under investigation in an effort to predict which patients are most likely to respond to immunotherapy. There are several types of biomarkers that are being investigated for their potential use in predicting response to immunotherapy in HNSCC patients, including tumor mutational burden, PD-L1 expression, tumor-infiltrating lymphocytes, immune gene structures, and circulating tumor DNA [63]. Such biomarkers can help select patients who are most likely to benefit from this treatment and avoid unnecessary toxicity in patients who are unlikely to benefit. Biomarkers may also help guide the selection of therapeutic nanoparticles that share a specific target. This emerging area of research is referred to as theranostics. Theranostic agents such as Lu-177 and nanoparticles are being studied in prostate and lung cancer [64,65]. Biomarker and theranostic research for HNSCC is still in its early stages, and much more work needs to be done before these markers can be used to predict response in clinical practice. 

Finally, there is a need for long-term follow-up studies to evaluate the efficacy and toxicity of radioimmunotherapy for HNSCC. This can help determine the durability of response and potential long-term toxicities associated with this treatment.

## 5. Conclusions

In conclusion, the combination of radiotherapy and immunotherapy is a potential treatment strategy for patients with LA and recurrent/metastatic head and neck squamous cell carcinoma (HNSCC). Clinical studies have explored the synergistic effects of these treatments, with several phase II/III trials indicating the safety and potential efficacy of adding immunotherapy to radiation. However, further research is required to optimize the use of radioimmunotherapy and identify the patients who are most likely to benefit. Ongoing clinical trials are investigating different immunotherapeutic agents, treatment sequencing, and optimal radiation techniques to guide decision-making in HNSCC treatment. De-escalation strategies, such as alternative fractionation and hypofractionation, have also emerged to balance effective tumor control with reduced treatment-related toxicity. Biomarker research is underway to identify predictive markers for immunotherapy response, but more extensive investigation is needed. Long-term follow-up studies are necessary to assess treatment durability and long-term toxicities associated with radioimmunotherapy in HNSCC. Through continued research and the refinement of protocols, radioimmunotherapy holds the potential to improve outcomes for patients with HNSCC, while minimizing treatment-related side effects.

## Figures and Tables

**Table 1 biomedicines-11-02097-t001:** Selected multi-institutional, phase III clinical trials for definitive IO in HNSCC.

Trial	Est. No.	Arms	Primary Endpoint	Sequence	Est. Completion
JAVELIN Head and Neck 100 [35]	697	CRT +/− avelumab	PFS	IO prior to, concurrent to, and after RT: lead-in phase IO, then concurrent CRT (cisplatin) with IO, then maintenance IO	Median PFS was not reached in either arms, but favors the placebo group.
GORTEC-REACH [36]	707	Definitive concurrent CRT +/− avelumab	PFS	IO concurrent with and after CRT (cisplatin or cetuximab): CRT with concurrent IO, then maintenance IO	For the cisplatin fit cohort, PFS HR was 1.27 (95% CI, 0.83–1.93). For cisplatin unfit cohort, PFS HR at 2 years was 0.85 (*p* = 0.15)
NRG-HN005 [37]	711	CRT (6 fx/week for 6 weeks) vs. CRT (5 fx/week for 6 weeks) vs. CRT (6 fx/week for 5 weeks) + nivolumab	PFS, QoL	IO concurrent with RT: concurrent CRT (cisplatin) +/− nivolumab	2/2025
KEYNOTE-412 [38]	804	CRT +/− priming, concurrent, and maintenance IO	EFS	IO prior to, concurrent to, and after RT: Priming IO, then concurrent CRT (cisplatin) with IO, then maintenance IO	6/2023
EA3161 [39]	636	Concurrent CRT +/− nivolumab vs. nivolumab alone	OS. PFS	IO after RT or no RT at all: concurrent CRT (cisplatin), then maintenance nivolumab or observation; vs. nivolumab alone	1/2027

CRT—chemoradiotherapy, PFS—progression-free survival, EFS—event-free survival, HNSCC—head and neck squamous cell carcinoma, IO—immunotherapy, RT—radiotherapy, fx—fractions.

**Table 2 biomedicines-11-02097-t002:** Selected multi-institutional, phase II/III clinical trials for pre-operative adjuvant radioimmunotherapy in HNSCC.

Trial	Est. No.	Arms	Primary Endpoint	Sequence	Est. Completion
IMvoke010 [40]	406	Atezolizumab vs. placebo	EFS	IO after: definitive local therapy, then IO	6/2027
NIVOPOSTOP [41]	680	Post-op concurrent CRT +/− nivolumab	DFS	IO prior to, concurrent with and after CRT (cisplatin): surgical resection, then adjuvant IO alone, then CRT with concurrent IO, then maintenance IO	9/2027
RTOG 1216 [42]	613	RT + cisplatin vs. RT + docetaxel vs. RT + docetaxel + cetuximab vs. RT + cisplatin + atezolizumab	DFS, OS	IO concurrent to RT	1/2027

CRT—chemoradiotherapy, DFS—disease-free survival, EFS—event-free survival, HNSCC—head and neck squamous cell carcinoma, IO—immunotherapy, RT—radiotherapy.

**Table 3 biomedicines-11-02097-t003:** Selected multi-institutional, phase II/III clinical trials for pre-operative neoadjuvant radioimmunotherapy in HNSCC.

Trial	Est. No.	Arms	Primary Endpoint	Sequence	Est. Completion
KEYNOTE-689 [46]	704	CRT +/− neoadjuvant and adjuvant pembrolizumab	mPR, EFS	IO prior to and concurrent with RT: neoadjuvant pembrolizumab, then surgical resection, then CRT (cisplatin) with adjuvant pembrolizumab for high risk patients or RT with adjuvant pembrolizumab for low risk patients	7/2026
IMSTAR-HN [47]	276	CRT +/− neoadjuvant nivolumab with adjuvant IO (nivolumab +/− ipilimumab)	DFS	IO prior to and after RT: Neoadjuvant nivolumab, then surgical resection, then adjuvant RT or CRT (cisplatin), then maintenance nivolumamb +/− ipilimumab	5/2024
CompARE [48]	785	CRT +/− neck dissection as indicated vs. induction IO, then CRT +/− neck dissection as indicated, then IO	EFS, OS	IO prior to or after RT	12/2026

CRT—chemoradiotherapy, DFS—disease-free survival, EFS—event-free survival, HNSCC—head and neck squamous cell carcinoma, IO—immunotherapy, RT—radiotherapy, mPR—major pathological response.

## Data Availability

No new data were created or analyzed in this study. Data sharing is not applicable to this article.

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
