# Peer review of "Combining Radiotherapy and Immunotherapy in Head and Neck Cancer"

_biomedicines, 2023, doi:10.3390/biomedicines11082097_

Round 1

Reviewer 1 Report

In their review manuscript entitled, “Radioimmunotherapy in Head and Neck Cancer”, Runnels et al summarize recent data on immunotherapy for HNSCC and discuss the potential synergy of radiation therapy and immunotherapy as treatment modalities, including in the adjuvant and neoadjuvant settings. The topic is important, as outcomes in HNSCC remain poor and immunotherapy has failed to deliver the spectacular results it has produced in other cancers.

On the whole, the manuscript is very difficult to read, as the authors examine and compare multiple clinical trials arms design in great detail, but provide little overarching summaries. There are unexplained abbreviations (e.g. LA, SoC), there is no easily accessible list - or description - of biologics mentioned (e.g. the authors list immunotherapeutics pembrolizumab, nivolumab, durvalumab, avelumab, atezolizumab, tremelimumab, camrelizumab, ipilimumab, as well as cetuximab – without mentioning the respective molecular targets); the description of the preclinical and mechanistic data is limited and not always accurate.

My main suggestions are, therefore, as follows: Expand existing tables to accommodate clinical trial arms design (information in lines 270-287 and 298-316, should also be transferred to the table format); in the main text, instead provide accessible summaries of the results and where relevant, pair these with supporting mechanistic and preclinical data; list biologics in an easy reference table, or alternatively precede every mention with the target (e.g. anti-PD-L1 antibody avelumab). This should make the manuscript more readable.

Specific points.

The title: “Radioimmunotherapy” is not an accepted term; Combination of radiation therapy and immunotherapy…. would be preferable

Line 32, “While some subsites of disease, such as human papilloma virus (HPV) mediated oropharyngeal cancer,” – “subsite” is an anatomical term, so “subset” or similar would be preferable here

Line 330, ‘There are several types of biomarkers that are being investigated for their potential use in predicting response to immunotherapy in HNSCC patients, including tumor mutational burden, PD-L1 expression, tumor-infiltrating lymphocytes, immune gene structures, and circulating tumor DNA.”

This information is important and should be provided upfront, and where possible the results should be discussed in view of these response biomarkers (e.g. 048 had 3 arms: aPD1 alone,  aPD1+chemo and aEGFR+chemo, and while groups were randomised irrespective of the PDL1 status,  aPD1 monotherapy had the most benefit  in PDL1 positive cancer, thus interpretation in line 77 is much more complex).

Line 54, “Cancer cells are constantly mutating with select clones developing the ability to evade immune-mediated recognition and destruction” – lack of immune recognition is not always the consequence of escalating immune evasion through the accumulation of mutations, as some cancers are immunogenic and others are not

Line 58, “Immune checkpoints keep immunologic cells such as T-cells in an inactive state;” – dysfunctional state is not the same as inactive/naïve state

Line 74, “Both trials found an improvement in overall survival (OS)” – please provide numbers

Line 82, ‘HNSCC patients with recurrent, unresectable or metastatic” – please finish the sentence

Line 94, “several studies have looked at the interaction between the HPV status and the effect of immunotherapy” – did the authors mean a correlation?

Line 95, does this refer to HPV or tobacco? Please explain

Line 108, “immunotherapy and radiation therapy may have a synergistic relation in this setting” – please explain which setting.

Line 118, “Choice of IO agents currently being tested include drugs targeting both PD-1/L1 …” – do the authors mean, concurrently?

Line 130, 2.1 Definitive – please expand subtitle

Line 132, extra comma

Table 1, SoC – please explain

Line 158, “Standard of care was….cetuximab radiation” – please correct treatment description

Line 166, “Subset analysis of PD-L1 high expression (CPS≥1) ..” – please reconcile with 25% on JAVELIN

Line 171, please provide a summary for 2.1

Line 173, 2.2 Per-operative: Adjuvant - by definition, adjuvant setting is post-operative

Line 175, “no initial results have yet to be reported’ – please correct

Line 189, “2.3 Peri-operative: Neoadjuvant  ‘ – by definition, Neoadjuvant setting is pre-operative

Line 194, “syngeneic murine oral cavity carcinoma models” – these were SC transplantable models. Are there any other relevant preclinical studies?

Line 197,” LA HNSCC” – please explain LA

Line 208, “dose escalated chemoradiation (64 Gy in 25 Fractions + Cisplatin)” – please explain dose escalation

Line 210, “All arms will be following by +/- neck dissection.” – did the authors mean to say that tumor resection will be performed for all study arms?

Line 211 is a repeat of 191. “IO for LA HNSCC clearly offers several advantages, including the opportunity to "debulk" tumors before surgical resection” – this is assuming that tumors do actually respond to treatment, what is the supporting evidence?

Line 241, “preclinical” - please expand and include the references

Line 229, “While several mechanistic rationales for the combination of these treatments exist,” – please list/explain the rationale(s)

4. Discussion – please remove trial design descriptions (270-287 and 300-316) to tables, and provide summaries instead

Line 335, “Biomarker research on immunotherapy for HNSCC is still in its early stages” – what about lessons from other cancers?

Lie 343, “In conclusion, the combination of radiotherapy and immunotherapy has demonstrated great promise as a treatment strategy for patients with locally advanced and recurrent/metastatic head and neck squamous cell carcinoma (HNSCC).” – this does not follow from the manuscript, as the results discussed were largely negative or not yet available for interpretation

English language is fine; however, typos need to be corrected, shorthand re-written in standard English and sentences finished where relevant.

Reviewer 2 Report

This review article is written very well. It's an original work and contributes to the field. Nevertheless, it's interesting to see the outcomes from NRG_HN005, KETNOTE-412, and EA3161 trials indicated in Table 1 at the end of the completion date. 

Author Response

Point 1: This review article is written very well. It's an original work and contributes to the field. Nevertheless, it's interesting to see the outcomes from NRG_HN005, KETNOTE-412, and EA3161 trials indicated in Table 1 at the end of the completion date.

Response 1: Thank you for your thoughtful feedback and for taking the time to review our manuscript. We certainly agree that the outcomes of NRG_HN005, KEYNOTE-412, and EA3161 will contribute substantially to present knowledge. We have verified that results from these trials have not yet been published.  

Reviewer 3 Report

Your review is well-written with updated informations on the therapy of head and neck squamous cell carcinoma. It would be better to make additional statements on abscopal effect of radiotherapy, internal radiation using theranostic agents such as Lu-177  peptides like HN-1 or nanoparticles.
